# Temperature but Not Photoperiod Can Predict Development and Survival of an Invasive Apple Pest

**DOI:** 10.3390/insects14060498

**Published:** 2023-05-29

**Authors:** Xiong Z. He, Qiao Wang

**Affiliations:** School of Agriculture and Environment, Massey University, Private Bag 11222, Palmerston North 4414, New Zealand; x.z.he@massey.ac.nz

**Keywords:** biosecurity, temperature, photoperiod, phenology, pest risk analysis

## Abstract

**Simple Summary:**

The apple leaf-curling midge is an invasive pest of apple. The pest can pupate on fruit and contaminate fresh fruit for export, causing biosecurity problems to many uninvaded regions such as Australia, California, China, India, Japan, and Taiwan. To generate critical knowledge for effective pest risk analysis, forecast and control of this pest, we tested the effects of five temperatures and five daylengths on its development and survival and developed a thermal model for the prediction of the number of generations per year under different climate conditions. We show that temperature but not daylength influenced its development and survival. The lower pupation and emergence rates at ≥25 °C may reduce the probability of its population build-up in regions where the average maximum temperature during the summer is over 25 °C. The model developed in this study can accurately predict the number of generations per year and the time of adult emergence in each generation under different thermal locations.

**Abstract:**

The apple leaf-curling midge, *Dasineura mali* Kieffer (Diptera: Cecidomyiidae), is an invasive pest of apple, and can contaminate fresh fruit for export, causing biosecurity problems. To provide crucial information for its pest risk analysis, forecast, and management, we investigated the effects of temperatures (5, 10, 15, 20 and 25 °C) and daylengths (10, 11, 12, 13, 14 and 15 h) on its development and survival. The midge eggs failed to hatch at 5 °C and larvae could not complete development at 10 °C. Pupation and emergence rates were significantly higher at 20 °C than at 15 °C and 25 °C. Daylength had no effect on these parameters. The low temperature threshold and thermal requirement to complete development from eggs to adults were 3.7 °C and 627 degree-days, respectively. The midge had a significantly lower thermal requirement for the completion of its lifecycle at 20 °C (614.5 degree-days) than at 15 °C (650.1 degree-days) and 25 °C (634.8 degree-days). The thermal model developed in this study provided accurate predictions of the number of *D. mali* generations and adult emergence time in each generation in different regions of New Zealand. We suggest that the model could be used to predict population dynamics of this pest in other parts of the world.

## 1. Introduction 

The apple leaf-curling midge, *Dasineura mali* Kieffer (Diptera: Cecidomyiidae), native to Europe, is a monophagous herbivore that attacks apple trees (*Malus* spp.) in Europe, South and North America, and New Zealand [1,2,3,4,5]. Its larvae feed on young leaves and shoots, reducing photosynthesis, resulting in poor shoot development, and potentially affecting the long-term yield [6]. Although the midge does not feed on fruit, its cocooned larvae can contaminate fresh fruit, causing quarantine problems [7,8]. *Dasineura mali* infestations in commercial orchards have raised concerns in the apple industry in New Zealand and some regions that trade with New Zealand, such as Australia, California, China, India, Japan, and Taiwan [9,10].

*Dasineura mali* adults emerge from the soil in the early morning and swarm around the emergence sites for mating, after which time females lay their eggs on growing apple shoots [11]. Eggs hatch in 3–5 days depending on the temperature and the larval feeding on young leaves causes them to roll or produce galls alongside their outer edges; mature larvae (3rd instar) leave the curled leaves and burrow in the soil (≤10 mm in depth) where they spin cocoons and pupate [12] but some can spin cocoons on fruit on their way to the ground [9,13]. The number of generations of *D. mali* per year varies depending on climate conditions, particularly temperature and availability of food, for example, there are three generations in Canada [5,14], three to six generations in Europe [15] and four to five generations in New Zealand [12,16,17,18].

Knowledge on the development, survival, and emergence of an insect species under different environmental conditions is vital to its pest risk analysis, forecast and management. Temperature is a main environmental factor significantly influencing these biological parameters [19]. Under optimal thermal conditions insects can complete their development with a high survival rate, but under extreme conditions such as high or low temperatures, they can have a high mortality rate and fail to complete a lifecycle [19,20]. The low temperature threshold (temperature below which no measurable development takes place) and thermal constant (the number of degree-days above the lower threshold temperature for the completion of development) are the two most important thermal parameters that determine how the temperature affects the developmental rate of ectotherms [5,21] and help predict pest phenology [22,23]. Some studies show that both temperature and photoperiod affect survival and development of cecidomyiid species [24,25,26,27,28,29,30,31,32,33,34,35,36]. 

Daylength is also a main environmental factor that can affect insect development, survival, and emergence [19]. However, photoperiod alone rarely affects the rate of development in insects [37,38]. For example, daylength does not appear to have any effect on development and survival in several dipteran species [27,30,38]. Although photoperiod is the most reliable cue of seasonal change in temperate regions, which may trigger diapause in some insects as unfavorable conditions approach [19,37,39], no diapause has been observed in *D. mali*, suggesting that daylength may not affect its developmental rate. Yet, how temperature and daylength influence survival and development of this midge pest have not been experimentally tested, making its risk analysis, forecast, and control difficult.

To generate critical information for the development of measures for effective pest risk analysis, forecast, and control of *D. mali*, we carried out experiments under a series of temperature and daylength conditions. We tested how temperature and daylength affected the development and survival of this pest. Based on our findings, we then calculated developmental thresholds and thermal requirements and developed a model for the prediction of the number of *D. mali* generations per year under different climate conditions in New Zealand. Our findings would help countries that trade with *D. mali*-infested regions perform pest risk analysis and develop forecast and management measures.

## 2. Materials and Methods

### 2.1. Insects 

We established a breeding colony in the laboratory from the field-collected mature larvae (3rd instar) in a mature organic apple orchard in Plant Growth Unit, Massey University, Palmerston North, New Zealand. The larvae were transferred to the rearing medium (vermiculite) in Petri dishes (5.5 cm in diameter × 1.3 cm in height) and maintained at 20 ± 1 °C, 65 ± 5% RH and 15:9 h (light:dark) for pupation. Newly emerged adults were released into an aluminium-framed experimental cage (43 × 42 × 40 cm) in which 10 apple seedlings (≈20 cm height, bred from rootstock MM 106 FSV) were maintained for oviposition. The cage had a fine metal mesh (aperture size = 0.25 mm) on the back and both sides and Perspex on the top and front and aluminium alloy on the bottom.

### 2.2. Effect of Temperature on Development and Survival 

We investigated the effect of temperature on the development, survival, and emergence of *D. mali* at five constant temperatures (treatments): 5, 10, 15, 20 and 25 °C with a day length of 15 h in the laboratory. For each test temperature, at the onset of photophase we released 30 newly emerged *D. mali* adults from the above colony with a sex ratio of about 1:1 into an above-mentioned experimental cage for oviposition, where 10 apple seedlings (replicates) were maintained as above. We removed seedlings 12 h after oviposition, caged them individually using transparent permeable plastic bags (42 cm in length × 23 cm in width with aperture = 0.25 mm) and maintained them at the test temperature. Because midge adults laid few eggs at 5 and 10 °C, we allowed adults to lay eggs on seedlings at 20 °C for 12 h and then individually caged the oviposited seedlings as above and transferred them to these two temperature conditions. 

The number of eggs laid on each seedling was counted under a stereomicroscope (Olympus, Tokyo, Japan). To determine the egg incubation duration and hatch rate, we made daily examinations of two infested shoots at each temperature under the stereomicroscope. All seedlings were maintained at the same temperature until larvae became mature (3rd instar, orange in colour). We started examining larval maturation by gently opening the curled leaves when they turned brown. We counted and removed mature larvae from leaves once a day until all larvae became mature. Egg hatch rate (no. of newly hatched larvae/no. of eggs) and larval survival rate (no. of mature larvae/no. of newly hatched larvae) were calculated. Developmental duration of eggs and immature larvae were recorded.

Because no eggs hatched at 5°C and no larvae completed development at 10 °C, we collected mature larvae from 10 seedlings under each of the rest of the test temperatures (15, 20 or 25 °C) and transferred them into Petri dishes with the rearing medium as mentioned above and maintained at the same temperature until adult emergence. There were 18, 20, and 20 dishes with 20, 30, and 25 larvae per dish established at 15, 20, and 25 °C, respectively. Newly emerged adults were counted and sexed daily according to Gagné and Harris [40]. After all adults had emerged, midge cocoons from each dish were examined under the microscope to determine the pupation rate of larvae (no. of pupae/no. of mature larvae) and the emergence rate of adults from the pupae (no. of emerged adults/no. of pupae).

### 2.3. Effect of Daylength on Development and Survival 

In this experiment we tested the effect of daylength on the development, survival, and emergence of *D. mali* under six daylength conditions: 10, 11, 12, 13, 14, and 15 h light based on a 24-h cycle at 20 °C in the laboratory. This temperature is the optimal thermal condition for *D. mali* development (see Section 3). The experiment was carried out and data were recorded as in the temperature experiment. There were 13, 17, 15, 25, 25, and 20 dishes with 25 mature larvae per dish at above daylengths, respectively.

### 2.4. Low Temperature Thresholds and Thermal Constants 

Based on the developmental time at test temperatures, we used the method of Campbell et al. [41] to estimate the low temperature threshold (*t*) and thermal constant (degree-days, *K*) for *D. mali*: *r* = *a* + *bT*, where *r* is the developmental rate [1/developmental duration (days)], *T* is the temperature, and *a* and *b* are estimates of the *r* intercept and slope, respectively. The low temperature threshold and thermal constant were calculated as *t* = −*a*/*b* and *K* = 1/*b*, respectively. The standard errors of *t* and *K* were calculated as $SE_{t}=\frac{\bar{r}}{b}\sqrt{\frac{s^{2}}{N\times \bar{r^{2}}}+{\left[\frac{SE\left(b\right)}{b}\right]}^{2}}$ and $SE_{k}=\frac{SE\left(b\right)}{b^{2}}$, respectively, where *SE*(*b*) is the *SE* of *b*, *s*^2^ is the residual mean square of *r*, $\bar{r}$ is the sample mean, and *N* is the sample size. At a given constant temperature, the thermal requirement for *D. mali* to complete development from eggs to adults was calculated as: *K* = *n* × (*T* − *t*), where *n* is the number of days required to complete development.

### 2.5. Estimation of the Number of Generations in the Field

Based on the low temperature threshold and degree-days (see Section 3), we estimated the number of generations ALCM might have in four regions in New Zealand, Hawke’s Bay (latitude 39.6° S), Palmerston North (latitude 40.4° S), Nelson (latitude 41.3° S) and Central Otago (latitude 45.3° S), by developing a degree-day model [22]: $N=\left(\frac{1}{K}\right)\overset{i=n}{\underset{i=1}{\sum }}\left(T_{i}-t\right)$, where *T_i_* is the mean daily air temperature (1981–2010) [42] at day *i* after the midge emergence from the overwintered generation, and *n* is the number of days during the growing season. To avoid overestimation or underestimation, we calculated the degree-days since the mean date of the first adult emergence peak, i.e., 30 September (2004–2017) in Hawker’s Bay, 2 October (2004–2017) in Nelson and 11 October (2001–2017) in Central Otago [18] and 10 October (2005–2007) in Palmerston North [43]. Hawke’s Bay, Nelson, and Central Otago are the three main apple growing regions for export in New Zealand [44].

### 2.6. Statistical Analysis

We performed analyses using SAS v. 9.4 software (SAS Institute Inc., Cary, NC, USA). Data on the development of different stages at different temperatures and daylengths and the thermal requirement (degree-days) were not normally distributed (Shapiro-Wilk test, UNIVARIATE Procedure) and thus analyzed using a generalized linear model (GLMMIX Procedure) followed by a Tukey-Kramer test for multiple comparisons between temperature. The relationship between the developmental rate and temperature was estimated using a general linear model (GLM Procedure). The survival data were analyzed using a generalized linear model (GENMOD Procedure) with a Binomial distribution and logit function. Multiple comparisons between temperatures or between day lengths were performed using the Contrast statement.

## 3. Results

### 3.1. Effect of Temperature on Development and Survival 

Eggs failed to hatch at 5 °C. At 10 °C eggs hatched but larvae failed to complete their development. The midge developed significantly faster with the increase of temperature (*F*_3,127_ = 70.72 for egg, *F*_2,993_ = 379.87 for larva, *F*_2,993_ = 812.80 for larva-adult, *F*_2,993_ = 1250.65 for total, *F*_2,470_ = 586.28 for male and *F*_2,523_ = 664.01 for female; *p* < 0.0001) (Table 1).

Egg hatch and larval survival rates were higher at 20 °C but there was no significant difference between temperatures ($x_{3}^{2}$ = 6.59, *p* = 0.0863 for egg hatch rate; $x_{2}^{2}$ = 3.49, *p* = 0.1747 for larval survival rates) (Figure 1). Significantly more mature larvae pupated and adults emerged at 20 °C than at 15 and 25 °C ($x_{2}^{2}$ = 32.29 and 23.59 for pupation and emergence rates, respectively; *p* < 0.0001) (Figure 1).

### 3.2. Effect of Daylength on Development and Survival 

Our results show that daylength has no significant effect on the development of *D. mali* (*F*_5,231_ = 0.53 for egg, *F*_5,207_ = 0.18 for larva, *F*_5,1743_ = 1.55 for larva-adult, *F*_5,1743_ = 0.77 for complete lifecycle, *F*_5,806_ = 0.42 for male and *F*_5,937_ = 0.59 for female; *p* > 0.05) (Table 2). Similarly, daylength had no effect on the midge survival ($x_{2}^{2}$ = 2.12, 7.18, 4.66 and 5.98 for egg hatch, larval survival, pupation, and emergence rates, respectively, *p* > 0.05) (Figure 2).

### 3.3. Low Temperature Thresholds and Thermal Constants

The low temperature threshold was 6.7 °C for egg development, which was twice as high as that for larval (3.2 °C) and pupal (3.4 °C) development (Figure 3A–C). Thermal requirement for *D. mali* developing from eggs to adults was about 630 degree-days with a low temperature threshold of about 3.7 °C (Figure 3D–F). Thermal requirement for the midge to complete development from eggs to adults was significantly different between three test temperatures with the lowest thermal constant being at 20 °C (ANOVA: *F*_2,993_ = 151.09, *p* < 0.0001) (Figure 4).

### 3.4. Estimation of the Number of Generations in the Field

Our model predicts that *D. mali* should have five generations a year in Hawker’s Bay with four completed generations between October and late March/early May, and mature larvae produced by the 4th generation would enter overwintering during late April (Figure 5A). However, four generations should occur in Palmerston North, Nelson, and Central Otago, with three completed generations during the growing seasons (October–mid-March) and mature larvae made by adults of the 3rd generation entering overwintering during mid-April (Figure 5B–D). 

## 4. Discussion

The midge had a significantly higher thermal requirement for the completion of its lifecycle at 25 °C than at 20 °C (Figure 4), suggesting that the temperature ≥ 25 °C may be less suitable for its development and explaining its summer outbreaks in New Zealand where the average maximum temperature rarely exceeds 25 °C [45]. Furthermore, the significantly lower pupation and emergence rates at 25 °C (Figure 1) or higher [46] may lower the probability of its population build-up in regions where the average maximum temperature during the summer is over 25 °C. The susceptibility to higher temperatures has also been reported in other cecidomyiid species such as the sorghum midge, *Contarinia sorghicola* (Coquillett) [25], and blackcurrant leaf midge, *D. tetensi* (Rübsaamen) [47]. These findings appear to match the fact that many plant-feeding cecidomyiid species are distributed in temperate regions [48]. 

The low temperature threshold of 6.7 °C for egg development in *D. mali* may explain why its eggs failed to hatch at 5 °C. The low temperature thresholds for the development of other *D. mali* stages were about 3.5 °C (Figure 3B–D), lower than the mean air temperatures during the winter in New Zealand [42], probably because this midge is native to colder Europe. Our results show that the low temperature threshold for larval development in *D. mali* was 3.2 °C (Figure 3B) but its larvae could not complete development at 10 °C. This is probably attributed to the fact that the higher mortality often occurs at lower temperatures due to the long developmental period [49]. Furthermore, larvae reduce their feeding activity at low temperatures [50], which may result in malnutrition and death. 

Using thermal parameters of several midge pests, various authors have developed models for the prediction of their population dynamics in the field, providing powerful tools for decision making in pest management [24,29,47,51,52,53]. For example, the accurate prediction of adult emergence time is critical to the applications of pheromone traps and insecticides [54,55,56,57]. Based on our findings in the present study (Figure 3D) and long-history climate data [40], we have developed a thermal model to predict the number of generations and the adult emergence time of each generation in four locations in New Zealand. We show that the model provides an accurate prediction of five generations in Hawker’s Bay and four generations in Palmerston North, Nelson, and Central Otago (Figure 5), agreeing to the field data reported [16,18,43]. Like other gall-forming insects [58,59,60], the availability of host plants for *D. mali* is also a crucial factor for the occurrence of the number of generations per year. For example, the lack of food in the autumn [16] may prevent it from having the fifth generation in Palmerston North, Nelson, and Central Otago.

Daylength is the most reliable cue of seasonal changes regulating temporal patterns of insect development, survival, and behavior [19,37,39], however its effect on insect life history traits may be species-specific. For example, studies on the sibling mosquitos, *Aedes aegypti* (Linnaeus) and *A. albopictus* (Skuse), reveal that daylength does not significantly affect the development and survival of the former species and the development of the latter species but the latter species has higher survival rate under short daylength [38]. In the present study, daylength had no effect on *D. mali* development (Table 2) and survival (Figure 2) under one temperature, 20 °C, which was optimal for the growth and development of the midge during the apple growing season. Our findings suggest that *D. mali* does not respond to daylength change and adjust its development and survival at the optimal temperature. 

However, both daylength and temperature change over the season in most parts of the world, particularly temperate regions. As a result, the effect of these environmental factors on developmental rate may be inter-dependent [61,62]. For example, the effect of short-day length on some midge species is associated with low temperatures [30,31,34]. In the predatory gall midge, *Feltiella acarisuga* (Vallot), photoperiod alone does not induce larval diapause, but diapause may occur when it feeds on the diapausing prey [29]. Although diapause has not been observed for *D. mali* anywhere in the world, it enters hibernation in winter due to the simultaneous decline of temperature, photoperiod, and food [16,43]. It may thus be worth investigating whether varying daylength and temperature would have combined effects on development, survival, and reproduction in this midge pest. 

In conclusion, we show that temperature significantly affects *D. mali* development and survival. We suggest that countries or regions with temperatures of ≥25 °C during the growing seasons may be less suitable for *D. mali* population build-up in apple orchards. Although we have not detected any effect of daylength on development and survival under one temperature, in future studies it may be worth testing whether associations of dynamic daylength and temperature would influence these parameters. The thermal model developed in this study can accurately predict the number of generations a year and the time of adult emergence in each generation in different locations of New Zealand. We recommend that this model be used to predict population dynamics of the pest in other parts of the world. Information generated in the present study can be used for pest risk analysis, forecast and control. 

## Figures and Tables

**Figure 1 insects-14-00498-f001:**
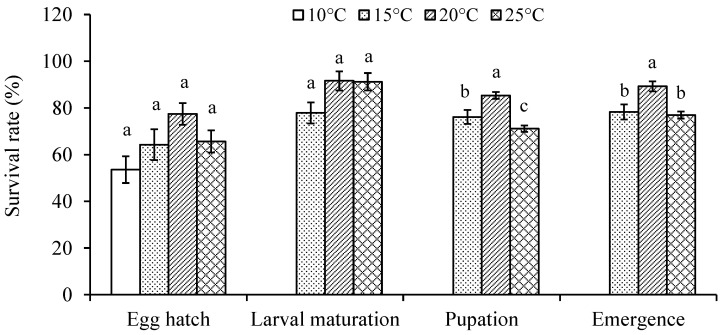
Mean (±SE) survival rate of four life stages in *Dasineura mali* at different temperatures. Columns with the same letters for each category are not significantly different (*p* > 0.05).

**Figure 2 insects-14-00498-f002:**
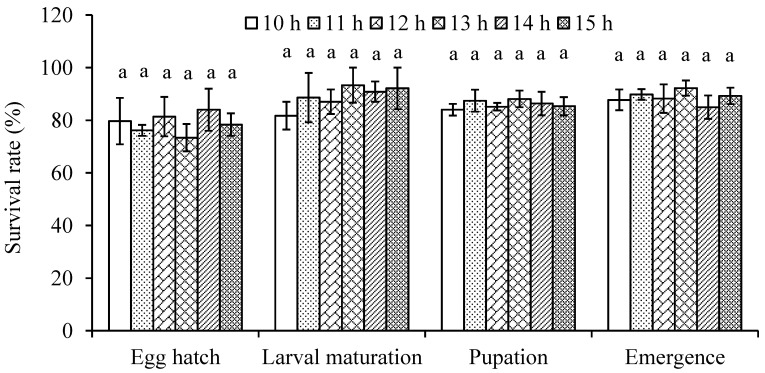
Mean (±SE) survival rate of four life stages in *Dasineura mali* at different daylengths ranging from 10 to 15 h at 20 °C. Columns with the same letters for each category are not significantly different (*p* > 0.05).

**Figure 3 insects-14-00498-f003:**
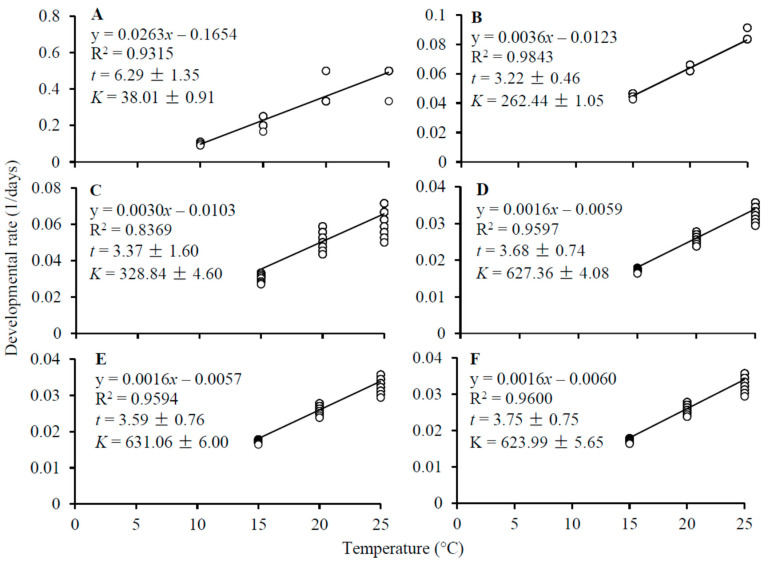
Linear regressions of developmental rate (r) on temperature (T), low temperature threshold (*t*) and thermal constant (*K*) for development of *Dasineura mali*: (**A**) egg, (**B**) larva, (**C**) mature larva-adult, (**D**) egg-adult, (**E**) male, and (**F**) female. All linear regressions are significant (*p* < 0.0001).

**Figure 4 insects-14-00498-f004:**
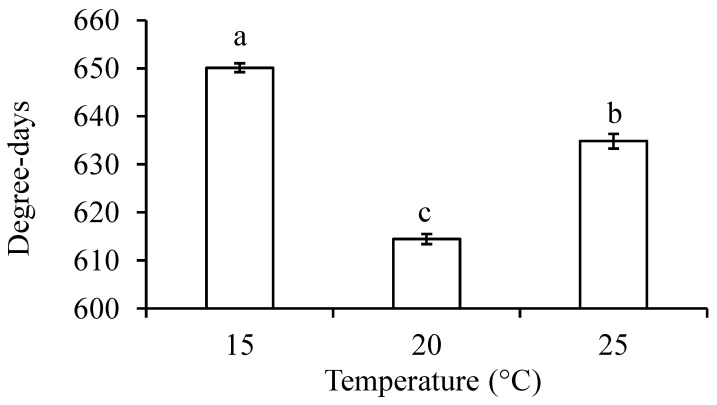
Thermal requirement for completion of development from eggs to adults in *Dasineura mali* at different temperatures. Columns (±SE) with the same letters are not significantly different (*p* > 0.05).

**Figure 5 insects-14-00498-f005:**
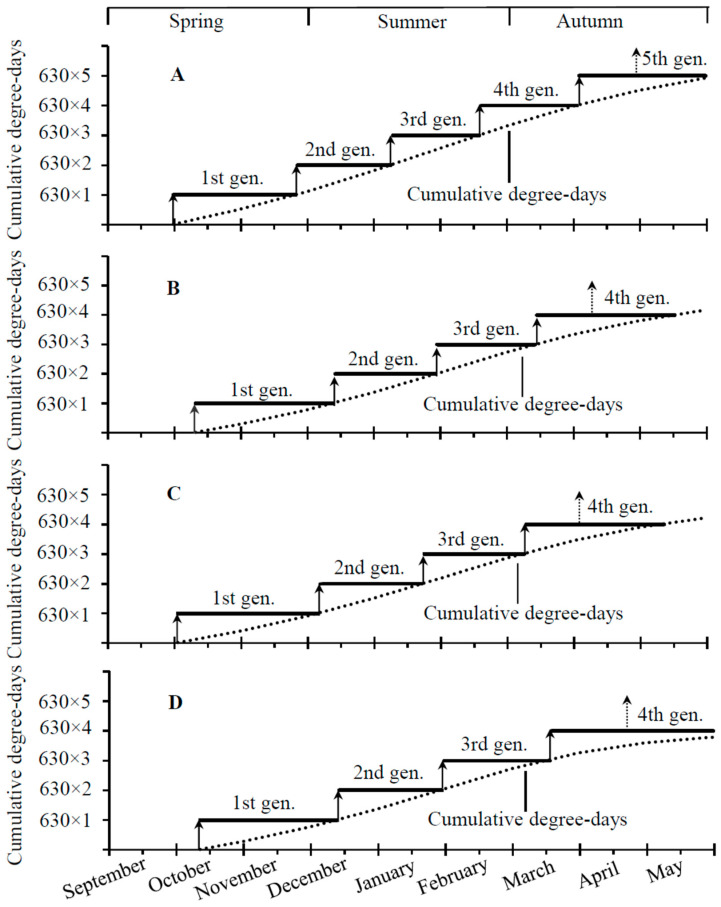
Predicted number of generations of *Dasineura mali* in different regions of New Zealand: (**A**) Hawker’s Bay, (**B**) Palmerston North, (**C**) Nelson, and (**D**) Central Otago. Arrows with a solid line indicate the time of adult emergence from the overwintered, 1st, 2nd, 3rd, and 4th generations, and arrows with a dotted line indicate the time of mature larvae entering overwintering.

**Table 1 insects-14-00498-t001:** Mean (±SE) developmental duration (days) of *Dasineura mali* at different temperatures under daylength of 15 h.

	10 °C ^1^	15 °C	20 °C	25 °C
Egg	9.89 ± 0.11 a	4.50 ± 0.10 b	2.85 ± 0.06 c	2.03 ± 0.04 d
Larva ^2^	-----	21.57 ± 0.02 a	15.99 ± 0.02 b	11.94 ± 0.01 c
Larva-adult ^3^	-----	31.41 ± 0.08 a	18.64 ± 0.06 b	15.77 ± 0.07 c
Total ^4^	-----	57.41 ± 0.80 a	37.63 ± 0.06 b	29.78 ± 0.07 c
Male ^4^	-----	57.38 ± 0.11 a	37.62 ± 0.09 b	29.81 ± 0.10 c
Female ^4^	-----	57.42 ± 0.12 a	37.65 ± 0.09 b	29.72 ± 0.10 c

Means followed by the same letters in row are not significantly different (*p* > 0.05). ^1^ Eggs and mature larvae (3rd instar) were transferred from 20 °C. ^2^ from 1st instar to 3rd instar. ^3^ from 3rd instar to adult. ^4^ from egg to adult.

**Table 2 insects-14-00498-t002:** Mean (±SE) developmental duration (days) of *Dasineura mali* at different daylengths at 20 °C.

	10 h	11 h	12 h	13 h	14 h	15 h
Egg	2.85 ± 0.06 a	2.92 ± 0.04 a	2.80 ± 0.06 a	2.84 ± 0.06 a	2.79 ± 0.06 a	2.85 ± 0.06 a
Larva ^1^	15.94 ± 0.06 a	15.91 ± 0.09 a	15.89 ± 0.10 a	15.91 ± 0.10 a	15.87 ± 0.09 a	15.97 ± 0.10 a
Larva-adult ^2^	18.83 ± 0.09 a	18.50 ± 0.06 a	18.24 ± 0.07 a	18.73 ± 0.11 a	19.18 ± 0.08 a	18.64 ± 0.06 a
Total ^3^	37.83 ± 0.09 a	37.49 ± 0.06 a	37.24 ± 0.07 a	37.73 ± 0.11 a	38.18 ± 0.08 a	37.64 ± 0.06 a
Male ^3^	38.00 ± 0.12 a	35.57 ± 0.09 a	37.28 ± 0.11 a	38.21 ± 0.14 a	38.14 ± 0.11 a	37.62 ± 0.09 a
Female ^3^	37.66 ± 0.12 a	37.45 ± 0.07 a	37.21 ± 0.10 a	37.15 ± 0.14 a	38.22 ± 0.13 a	37.65 ± 0.09 a

Means followed by the same letters in row are not significantly different (*p* > 0.05). ^1^ from 1st instar to 3rd instar. ^2^ from 3rd instar to adult. ^3^ from egg to adult.

## Data Availability

All data are included in the manuscript.

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
