# Peer review of "Temperature but Not Photoperiod Can Predict Development and Survival of an Invasive Apple Pest"

_insects, 2023, doi:10.3390/insects14060498_

Round 1

Reviewer 1 Report

The authors, collected mature larvae from 20ºC, and immediately transferred them to 10°C in 12 dishes to study the development at 10°C. Given that the eggs hatched at different temperature and were the transferred, I think this treatment should not be the part of analysis unless authors demonstrate that including it did not impact the overall analysis.

Also, in the results section, treatment degree of freedom and the error degree of freedom varies widely. Authors should provide little more description on number of replicates, treatments, dependent variables, and independent variables.

In the discussion section authors state that” Our results show that D. mali could not complete its lifecycle at 10°C but the high tolerance of its mature larvae and pupae to this temperature (Table 1) may allow it to survive lengthy shipment in cool containers and reach the countries that trade with midge-infested regions.” This statement appears to come from nowhere. Authors should provide background information about the apple trade practices, shipping conditions and potential of D. mali transportation during trade.

Author Response

Reviewer 1: The authors, collected mature larvae from 20ºC, and immediately transferred them to 10°C in 12 dishes to study the development at 10°C. Given that the eggs hatched at different temperature and were the transferred, I think this treatment should not be the part of analysis unless authors demonstrate that including it did not impact the overall analysis.

Our response: Thank you for your constructive comments, to which we agree. We have now deleted this treatment and relevant information in M&M, Results and Discussion. Following the above deletion, we have reanalysed our data. As a results, we have revised the Results section, including Table 1, Figure 1, and relevant statistic data in brackets. Reanalyses have not changed our conclusions.

Reviewer 1: Also, in the results section, treatment degree of freedom and the error degree of freedom varies widely. Authors should provide little more description on number of replicates, treatments, dependent variables, and independent variables.

Our response: Thank you for your suggestions. We have revised the M&M and Results accordingly. Temperature and daylength were the dependent variables and the midge life history traits were the independent variables in this study.

Reviewer 1: In the discussion section authors state that” Our results show that D. mali could not complete its lifecycle at 10°C but the high tolerance of its mature larvae and pupae to this temperature (Table 1) may allow it to survive lengthy shipment in cool containers and reach the countries that trade with midge-infested regions.” This statement appears to come from nowhere. Authors should provide background information about the apple trade practices, shipping conditions and potential of D. mali transportation during trade.

Our response: As stated above, we have deleted this treatment and made revisions in Simple Summary, M&M, Results and Discussion accordingly.

Reviewer 2 Report

The manuscript provides a degree-day model for calculating generation time for the apple leaf-curling midge. The results show that as many as 5 generations per year can develop in NZ. The model could be used in other countries where the pest occurs.  Day length had no effect on generation time. The manuscript is very well written, free of typos. The statistical analysis is proper. The conclusions follow logically from the results, which are very straight-forward.  The only suggestions I have are minor. In the introduction line 2 where it states that D. mali is monophagous on apple trees, I suggest the authors put in parentheses (Malus spp.).  In the abstract line 5 there is an awkward sentence caused by the word "transferred."  Change to "transfer" but you could rewrite to clarify the details and reasons for the transfer. 

Congratulations on a well designed study and well written manuscript  

Author Response

Reviewer 2: The manuscript provides a degree-day model for calculating generation time for the apple leaf-curling midge. The results show that as many as 5 generations per year can develop in NZ. The model could be used in other countries where the pest occurs.  Day length had no effect on generation time. The manuscript is very well written, free of typos. The statistical analysis is proper. The conclusions follow logically from the results, which are very straight-forward.  The only suggestions I have are minor. In the introduction line 2 where it states that D. mali is monophagous on apple trees, I suggest the authors put in parentheses (Malus spp.). 

Our response: We appreciate your positive comments on our manuscript. We have added this information in Introduction.

Reviewer 2: In the abstract line 5 there is an awkward sentence caused by the word "transferred."  Change to "transfer" but you could rewrite to clarify the details and reasons for the transfer. 

Our response: Good point. We have deleted the treatment from analysis according to reviewer 1. As a result, this sentence has been removed.

Reviewer 2: Congratulations on a well designed study and well written manuscript.  

Our response: We appreciate your comments.

Round 2

Reviewer 1 Report

The authors have significantly improved the MS.